# DNA Hypermethylation and a Specific Methylation Spectrum on the X Chromosome in Turner Syndrome as Determined by Nanopore Sequencing

**DOI:** 10.3390/jpm12060872

**Published:** 2022-05-26

**Authors:** Xin Fan, Beibei Zhang, Lijun Fan, Jiajia Chen, Chang Su, Bingyan Cao, Liya Wei, Miao Qin, Chunxiu Gong

**Affiliations:** 1Department of Endocrinology, Genetics and Metabolism, National Center for Children’s Health, Beijing Children’s Hospital, Capital Medical University, Beijing 100045, China; fanxin602@163.com (X.F.); aazhangbei@126.com (B.Z.); fanlijun0916@163.com (L.F.); chenjiaj2009@126.com (J.C.); changsulucky@yahoo.com (C.S.); caoby1982@163.com (B.C.); wly0830@sina.com (L.W.); oqinmiaoo@126.com (M.Q.); 2Department of Pediatric, The Second Affiliated Hospital of Guangxi Medical University, Nanning 530007, China

**Keywords:** Turner syndrome, methylation, nanopore sequencing

## Abstract

The molecular genetic mechanism of Turner syndrome (TS) still leaves much to be discovered. Methods: TS (45X0) patients and age-matched controls (46XX and 46XY) were selected. The nanopore sequencing combined with trio-whole exome sequencing (trio-WES) were used for the first time to investigate TS. Results: Thirteen TS (45X0) patients and eight controls were enrolled. Trio-WES analysis did not find any pathogenetic or likely pathogenic variants except X chromosome (chrX) deletion. The average methylation levels and patterns of chrX in 45X0 and 46XY were similar, and significantly higher than in 46XX (*p* = 2.22 × 10^−16^). Both hyper-methylation and hypo-methylation were detected in the CpG island (CGI), CGI_shore, promoter, genebody, and PAR1-region, while in the transposon element inactivation regions of the chrX and hypermethylation were predominant. A total of 125 differentially methylated genes were identified in 45X0 compared to 46XX, including 8 and 117 hypermethylated and hypomethylated genes, respectively, with the enrichment terms of mitophagy, regulation of DNA-binding transcription factor activity, etc. Conclusions: The results suggest that the methylation profile in patients with TS might be determined by the number of X chromosomes; the patterns of methylation in TS were precisely associated with the maintenance of genomic stability and improvement of gene expression. Differentially methylated genes/pathways might reveal the potential epigenetic modulation and lead to better understanding of TS.

## 1. Introduction

Turner Syndrome (TS) is the most common sex chromosome aneuploidy in humans [1]. Clinical features of TS include typical dysmorphic features, short stature, hypergonadotropic hypogonadism, and other medical conditions such as cardiovascular, renal, skeletal, endocrine, and metabolic abnormalities, as well as autoimmune diseases and hearing problems. Patients with TS display a large heterogeneity of clinical symptoms, and clinical features tend to be varied in patients with the same karyotypes. Highly variable phenotypes are related to the monosomy of chromosome X, the variant’s absence of genetic material on the X chromosome. Some believe all 45X0 individuals who survive are cryptic mosaic; mosaic in all organs or tissues might be contributed, but this is difficult to conclude in living patients [2]. Recent advances suggest that genomic and epigenetic changes may also play important roles in TS. The highly variable clinical features can also result from epigenetic changes in the genome, which may be due to genomic imbalance and global network effects of a reduced X chromosome dosage due to the absence of the sex chromosome [3]. Hence, there are several aspects of the genetics mechanisms behind TS that still need to be unraveled.

DNA methylation is an important process that is involved in many biological processes, including transcription regulation, chrX inactivation, genomic imprinting, transposon inactivation, embryonic development, and chromatin structure modification [4]. However, epigenetic effects have been poorly investigated in TS as most of these studies have focused on clinical, chromosomal, and genetic abnormalities. Using a novel cDNA based high throughput approach of assessing genome wide methylation, Rajpathak et al. examined the methylation landscape in human fibroblasts in 45X0 and 46XX individuals. Their results showed differences in the methylation of X linked genes in these two situations, and the methylation of several autosomal genes was affected in this X monosomy state [5].

Recently, Zhang et al. analyzed the gene expression patterns using RNA-seq and DNA methylation patterns with genome-wide targeted capture bisulfite sequencing from clinically well-characterized TS with sex-matched controls. They observed that the chrX methylation patterns in XX and XXY were similar and showed significant differences compared to X0 and XY [3]. However, there are several limitations associated with bisulfite sequencing with its short-read techniques and use of immunoprecipitation sequencing. The method converts unmethylated cytosines into uracil residues and then detects these by using next-generation sequencing, but the conversion efficiency is often limited, and short-read sequencing cannot always assay the repetitive genomic regions. Immunoprecipitation followed by short-read sequencing can detect DNA or RNA modifications in the genomic regions, but it cannot achieve this to a single-base resolution. However, third-generation sequencing technology (such as nanopore sequencing), in addition to the long-read advantage, can also directly detect DNA methylation modifications and achieve an average precision up to 0.99 [6].

In this study, nanopore sequencing was combined with next-generation sequencing (trio-WES) and used for the first time to investigate the genomic information of patients with TS. This provided a basis for understanding the mechanisms of TS at the genomic, mutation, and epigenetic levels. The unravelling of the related genes and pathways might further reveal the potential epigenetic modulation of DNA methylation in TS and explain the diversity of clinical manifestations of these individuals.

## 2. Materials and Methods

In order to investigate the genomic and epigenetic information of TS individuals, we performed the nanopore sequencing combined with trio-whole exome sequencing (trio-WES) on patients with TS (45X0) and controls.

This study was approved by the Ethics Committee of the Beijing Children’s Hospital, Capital Medical University, National Center for Children’s Health. All the patients and control subjects involved in the study were admitted to the out-patients clinic from January to December 2019. Written informed consent was obtained from each participant or their legal guardian.

Thirteen patients with TS, aged 3 through 12.9 years, with classical Turner (karyotype 45X0) and diverse clinical features were studied along with their parents. Eight healthy controls, 4 males (46XY) and 4 females (46XX), who were age matched were also enrolled. The clinical data from the patients were collected, and 3mL of peripheral blood samples from all participants were drawn and processed for subsequent whole exome and nanopore sequencing. Patients with TS and their parents first underwent whole exome sequencing, in order to identify candidate casual variations leading to TS or specific clinical features, and then nanopore sequencing techniques for the detection of DNA epigenetic modifications were performed in all 21 individuals (Figure 1).

### 2.1. Next Generation Sequencing 

DNA was extracted from all the samples (patients with TS and their parents). Extraction of DNA from peripheral blood by kit from TianGen Biotech Co., Ltd., Beijing, China. The concentration of genomic DNA in peripheral blood was >10 ng/uL, and the total amount of genomic DNA was >1200 ng, volume >30 mL. Library preparation: after ultrasonic fragmentation, terminal repair, junction connection, magnetic bead purification connection, PCR amplification, and magnetic bead purification PCR system, concentration (Qubit quantitative: 25 uL, >15ng/uL); distribution range of DNA fragments of the library: 300–600 bp (Agilent2100 detection peak 430 bp). Hybridization capture: the xGen **^®^**ExomeResearchPanelv1.0 capture probe of IDT Company was used to capture the whole exon group of humans. The xGen **^®^**ExomeResearchPanelv1.0 capture probe covers the coding region of 19,396 genes in the human genome and captures the target interval size 39 Mb. A total of 429,826 capture probes were designed and synthesized. The library met the following requirements: library concentration Qubit quantitative: 20 uL, >5 ng Uniul; effective molecular concentration q-PCP quantitative: 20 uL, >10 nM.

Then, trio-WES was performed by using an Illumina NovaSeq 6000 series sequencer (PE150), and more than 99% of the target sequences were sequenced. The sequencing process was performed by Beijing Chigene Translational Medicine Research Center Co., Ltd., 100875, Beijing, China. The mutation frequency of SNP was screened by using local and public databases, and the disease associated with the mutation was annotated in combination with family genetic model and biological significance. The paired-end reads were performed using Burrows–Wheeler Aligner (BWA) to the Ensemble GRCh37/hg19 reference genome. Base quality score recalibration together with SNP and short indel calling were conducted using GATK. The average depth of proband sequencing was 240×, the minimum depth is more than 160×, and the amount of data obtained was approximately 20 GB. The average depth of sequencing of the proband’s parents was 110×, the minimum depth is more than 80×, and the amount of data is about 9 GB. A total of 19,119 genes were analyzed through the self-developed human whole exome analysis process. The parental origin of the X chromosome was determined according to the SNP and STR polymorphic loci of patients and their parents. Based on the BAM after removal of duplications, the sequencing depth of the target region was counted, and the continuous exon results were connected into large fragment CNV results by using sample data calculations.

### 2.2. Nanopore Sequencing and Data Analysis

#### 2.2.1. Sample Preparation

DNA was extracted from the samples (patients with TS and controls) and then fragmented using a Megaruptor (30 KB in length). This was followed by enrichment (blue Pippin) and purification. For Nanopore sequencing, library preparations were completed using the ligation sequencing kit (Cat# SQK-LSK109, Oxford Nanopore Technologies, Oxford, UK). All procedures were preformed according to the manufacturer’s protocols. 

#### 2.2.2. Library Construction and Sequencing

The PromethION platform was used to construct a library. This was constructed by using a Ligation Sequencing Kit (SQK-LSK109) obtained from Oxford Nanopore Technologies (ONT), and the R9 of the PromethION Series (Model 48) sequencer was subsequently used for sequencing. The requirements of sequencing data were set as follows: (a) average sequencing depth >20× with no less than 99% coverage and no less than 95% for 10×, (b) the average length of the clean data reads was not less than 15KB and the average mass value was no less than 9, and (c) the comparison efficiency was >95%.

#### 2.2.3. Bioinformatics Analysis

ONT Guppy software was used for base calling and the Nano pack software package was used to filter the data obtained. Then, Minimap2 software was used to compare the third-generation sequenced reads to the reference genome (GRCh37/ HG19).

#### 2.2.4. Methylation Analysis

The genome-wide methylation levels for the ONT assembly were calculated using Nano polish version 0.11.1 (Nano polish, RRID:SCR_016157) with called_sites ≥10. The methyl Kit [7] software was used to extract the differentially methylated regions (DMRs) within the genomic regions through the comparative analysis of patient and control samples.

#### 2.2.5. Functional Enrichment Analysis

Metascape (http://metascape.org, accessed on 6 February 2022) was used to functionally annotate the DMGs Ref. [8] obtained. All the genes in the genome were used for the enrichment background. The terms with a *p*-value < 0.01, a minimum of 3 counts, and an enrichment factor >1.5 (which is the ratio between the observed counts and the counts expected by chance) were collected and grouped into clusters based on their membership similarities.

## 3. Results

### 3.1. Clinical Information

Thirteen patients with TS and eight controls were enrolled into this study. The mean age of the patients with TS was 8.4 ± 3.3 y. They had highly variable clinical phenotypes and different complications: cardiovascular system abnormalities (4/13); urinary system abnormalities (4/13); skeletal malformations (6/13); thyroid dysfunction (2/12); intellectual impairment (3/13); and hearing impairment (2/13). The detailed clinical information is shown in Appendix A. The eight heathy controls included four males and four females, and their mean age was 8.5 ± 2.3 y.

### 3.2. Origin of the X Chromosome and Casual Variants Analysis in TS

Trio-WES was performed on the 13 patients with TS (45X0). In addition to the complete absence of one X chromosome, Trio-WES and exome-based copy number variation (CNV) analyses were performed on the WES data. Except for deletions of X chromosome, we did not find any pathogenetic/likely pathogenic single-nucleotide variant (SNV), nor small insertion/deletion, nor micro-chromone copy number variant, especially when we double checked the variation related to typical clinical characteristics of TS, such as short stature, hypergonadotropic dysplasia, and cardiac and renal malformation. All the patients with TS had a complete absence of one of the two X chromosomes. The X chromosome of four patients were maternally derived, whereas the other nine were paternally derived.

### 3.3. Genome-Wide Methylation Analysis

#### 3.3.1. Overview of Genome-Wide Methylation in TS

The CpG island sequencing data of the 21 samples (13 TS and 8 controls) were quality controlled and filtered with the filter condition of the methyl_read_sites ≥ 10. There were 419,022 CpG islands after any redundancies were removed, and 902,529 CpGs were left on the X chromosome.

There was no significant difference in the autosomal chromosomes between the 13 TS and 8 healthy controls, but the methylation levels and patterns in both 45X0 and 46XY were similar and showed significant differences when compared to 46XX (*p* = 2.22 × 10^−16^, Figure 2B). In addition, the X chromosome of four patients (P2, P5, P7, and P13) were maternally derived, and the rest were paternally derived. As we can see in Figure 2A, red boxes represented all the TS patients. There was no significant difference in the methylation profiles on X chromosome between the two groups.

#### 3.3.2. DNA Methylation Profiling of Different Functional Regions on the X Chromosome in TS

In order to clarify the epigenetic mechanisms in patients with TS, the methylation levels of different genomic regions on the chrX of 45X0 patients were analyzed. The methylation landscape of the chrX in the TS cohort was also explored, and this was focused on CpG islands (CGI) and CGI_shore, the promoter and genebody, the transposon elements (TE), and the PAR1-region, as well as the regions associated with the X chromosome inactivation (XCI).

In the CGI and CGI_shore regions (Figure 3), the methylation levels in patients with TS were similar to those of 46XY, and these were different from those of 46XX. The average methylation level in TS was lower than 46XX. Hypomethylation existed in both the CGI and CGI_shore regions, while hypermethylation occurred only in the CGI_shore regions.

The methylation patterns of patients with TS were similar to those of 46XY (Figure 4). Both hypomethylation and hypermethylation occurred in the promoter and genebody regions, but hypomethylation was predominant only in the promoter regions, where the average methylation level was lower than 46XX. However, hypermethylation occurred mainly in the genebody regions, and the levels were higher than in 46XX controls.

Transposon elements (Tes) are mobile genomic sequences classified by their mode of propagation. These can be short or long interspersed repeat elements (SINE and LINE, respectively) as well as retrovirus-like elements with long terminal repeats (LTR). High levels of hypermethylation occurred in the LINES (L1/L2), SINEs (Alu/MIR), and the transposons ERV1, ERVL–MALr, ERVK, and ERVL, regions (Appendix A). The average methylation level of these regions was similar to that of 46XY, and it was significantly higher than that of 46XX controls. Hypermethylation may function to maintain genome stability by suppressing the activities of transposons.

Regarding XIST and JPX regions, the XCI-related and pseudo autosomal region (PAR) are specific X chromosome regions. The X inactive specific transcript (XIST) encodes specific transcripts which are inactivated on the X chromosome. JPX is a nonprotein-coding RNA transcribed from a gene within the X-inactivation center that participates in XCI [9]. The average methylation levels of the XIST and JPX regions in 45X0 individuals tend to be similar to those of 46XY and are significantly higher than 46XX controls (Figure 5A–D). In addition, hypermethylation was observed in the 2K upstream and downstream regions of the transcription start sites (TSSs) of XIST and JPX (Figure 5E–H).

The pseudo autosomal regions, PAR1 and PAR2, are short regions of homology between the mammalian X and Y chromosomes. As there were only a few valid methylation sites found in the PAR2 region, only the methylation levels of the PAR1 region were analyzed. A slightly higher level of methylation was observed in TS 46XY when compared with that of the controls (Appendix A).

#### 3.3.3. DMGs and Functional Enrichment Analysis

Furthermore, we performed a thorough differential methylation analysis among these three groups studied. A total of 125 differentially methylated genes (DMGs) were identified in 45X0 when compared to 46XX, including 8 and 117 which were significantly differential hypermethylated (X0/XX hyper) and hypomethylated (X0/XX hypo) genes, respectively. However, 129 DMGs were identified between male and female controls (46XY and 46XX), among which 8 were differentially hypermethylated genes (XY/XX hyper) and 121 were differentially hypomethylated genes (XY/XX hypo). 

Interestingly, only 3 and 16 significantly differentially hypermethylated and hypomethylated genes were identified in 45X0 when compared to 46XY controls, respectively (Figure 6). There were 106 common genes in X0/XX and XY/XX hypo, and 3 common genes in X0/XX and XY/XX hyper. The three genes were BCOR, AK2, and RP11-453F18_B.1 (Appendix A). This indicates that the relationship between the X chromosome number and DNA methylation (the activity of chrX) was significant and suggests that the activity of chrX in TS individuals was similar to that of the normal males.

The pathway and process enrichment analyses of the identified DMGs were performed using Metascape (http://metascape.org, accessed on 6 February 2022). As shown in Figure 7, the enriched terms included mitophagy, tubulin deacetylation, regulation of DNA-binding transcription factor activity and protein autophosphorylation. In addition, the most significantly enriched reactome gene sets were protein localization, RHO GTPase cycle, RNA polymerase II transcription initiation, and promoter clearance (Appendix A). The KEGG pathway that is related to basal transcription factors (such as ko03022) was also significantly enriched among the DMGs (Figure 7).

#### 3.3.4. Potential Candidate Genes Associated with TS Phenotypes

Several studies have predicted a list of candidate genes for TS based on their gene expression and DNA methylation levels. Therefore, we next focused on those genes for which differential methylation was detected, and those which were associated with differential expression in TS in previous studies. Here, we identified 130 hypomethylated genes (X0_Hypo), among which 68 were identified by Sharma et al. [10], and 81 were found by Zhang et al. [3] (Figure 8, Appendix A). In addition, we also identified 16 differentially hypermethylated genes, and observed three of them (BCOR, AK2, and RP11-453F18_B.1) were common in TS and 46XY when compared to 46XX. There were also three genes (HOXB3, HOXB-AS3, and DTX2) which were hypermethylated in TS when compared to 46XY.

## 4. Discussion

Recent studies suggest that multiple genetic conditions are involved in TS, including single gene variations, rare copy number variants, genetic polymorphisms, and epigenetics [1,11,12,13]. Although there are many different types of genetic tests, there is no single test that can detect all the different genetic conditions. The approach to genetic testing is usually based on the type of genetic condition a patient is being tested for. In this study, we selected the patients whose karyotype of white cells in peripheral blood was 45X0, and sought to exclude mosaicism from the general karyotype by counting metaphase chromosomes (≥50), which was the standard way. Then, nanopore sequencing was combined with trio-WES and used for the first time to investigate the genomic information of patients with TS; these were compared with control subjects. 

### 4.1. Genetic Information from Trio-WES

Initial analysis based on trio-WES indicated that the X chromones in nine patients with TS were maternally derived, whereas the other four were paternally derived, which is consistent with previous studies. In most TS cases, there is a loss of the paternal X chromone [14]. Trio-WES and exome-based copy number variant (CNV) analyses were performed on WES data. However, we did not find any pathogenetic or likely pathogenic single-nucleotide variant (SNV), or small insertion/deletion, or chromone copy number variant, except the deletion of the X chromosome. Additionally, we also paid special attention to those related to typical clinical characteristics of TS, such as short stature, hypergonadotropic dysplasia, cardiac and renal malformation, etc., but did not find any of these either. Trio-WES failed to detect other rare variants associated with specific clinical features in the TS individuals, except for X chromosome deletion. After excluding other rare variants, we conducted methylation analysis to explore the phenotypic heterogeneity of TS. As is now well known, DNA methylation has been increasingly associated with the disease state [15].

There is a hypothesis all the TS individuals alive were cryptic mosaic, and that the ratios of abnormal cells in different tissues of variant karyotype were not the same. Additionally, it is reported that 50% TS had ≥5% difference in the degree of mosaicism between the left- and right-hand sides of the buccal mucosa [16]. On the other hand, the proportion of 46XX cells increases with time in women with Turner syndrome, seen in a 10-year follow-up study [17]. Therefore, it is difficult to identify how whole organisms’ methylation in TS change by peripheral blood DNA. In this study, we selected deep trio-whole exome sequencing (average depth of sequencing is 240×, the minimum depth is more than 160×) in TS individuals, which contributed to recognize mosaic to some extent. However, the detection of mosaicism in human disease has been challenging. To date, we still cannot exactly conclude to what extent the patterns in peripheral blood can be generalized to whole organisms. In future study, analysis of tissue-specific profiles combined with peripheral blood profiles might be helpful to better understand.

### 4.2. Profiling Genome-Wide and chrX DNA Methylation in TS 

#### 4.2.1. Genome-Wide DNA Methylation in TS

In humans, DNA methylation occurs at the fifth position of the pyrimidine ring of the cytosine residues within the CpG sites to form 5-methylcytosines, and CpG methylation is common in humans. CGIs are regions of the genome that contain a large number of CpG dinucleotide repeats. Most of the DNA methylation in humans occurs on cytosines that either precede a guanine nucleotide or a CpG site. CGIs are highly concentrated parts of the genome, and these are usually in the gene promoter regions of chromosomes [18]. The remaining CpG sites are usually spread out across the genome. 

A previous study using the 450K-Illumina Infinium assay revealed that both autosomal and the X-chromosome showed hypomethylation at most of the differentially methylated positions. Additionally, genome wide methylation profiles can clearly distinguish TS from controls [19]. In this study, there was no difference in methylation levels of autosome chromosomes and genome-wide hypermethylation occurred in 45X0, which was different to previous studies [3,19]. We speculate that this may be related to different detection methods. Previously, direct identification of methylation from native DNA was not possible. For example, during the bisulfite treatment process in the traditional DNA methylation method, with genome-wide targeted capture bisulfite sequencing, which is the first step used to convert unmethylated cytosine residues to uracil, an increase in the incomplete chemical conversion and biases from sources tended to occur. However, nanopore sequencing works by monitoring changes in the electrical charges on the nucleic acids when these are passed through a protein nanopore [20]. The significant advantages of nanopores sequencing include no labeling, long-reading, and it is not usually affected by the amplification bias introduced by PCR. In addition, the number of CpG sites obtained is more than that of the traditional methylation detection method, therefore nanopore sequencing can faithfully estimate the DNA methylation landscape. However, nanopore sequencing does have its disadvantages. It tends to be error prone, with error rates of approximately 15%. This error rate is tolerable because the method sequences the same sequence several times.

#### 4.2.2. Profiling DNA Methylation on the X Chromosome

In previous studies, it was confirmed that the methylation profile might be determined by the number of X chromosomes, and whether these were hypomethylated in TS [3,19]. In 13 patients, the X chromosomes of four patients (P2, P5, P7, and P13) were maternally derived, and those of the other 9 patients were paternally derived; however, there was no significant difference in the methylation profiles between the two groups. As we can see in Figure 2A, red boxes represented all the TS patients. In this study, significantly elevated mean methylation levels occurred on the X chromosomes. Further, methylation characteristics of different functional regions in TS are the first to be reported in this study. Both hypermethylation and hypomethylation were detected in the CGI, CGI_shore, promoter, genebody, and PAR1-regions, while in the TEs, XCI, and JPX regions, hypermethylation was predominant. The trends of hypomethylated and hypermethylated regions on the PAR1-region in TS were more obvious than those observed in the controls.

In general, the higher the degree of methylation in the transcriptional regulatory regions, the greater the gene expression would be downregulated. Therefore, hypomethylation in the CGI and promoter regions were conducive to enhanced transcription. However, when away from the CGI regions, methylation levels can be gradually elevated, and this can also be conducive to transcription. According to a previous study, hypermethylation in the genebody region was positively correlated with transcription efficiency, which can be additionally conductive to transcription [19]. Therefore, it appears that all the methylation changes described above are beneficial to the improvement of gene transcription in TS.

TEs make up to two-thirds of the human genome and are known to play important roles in molecular functions [21]. TEs such as LINEs and SINEs have numerous effects on the genome, including genomic rearrangements and the regulation of gene [22]. In this study, hypermethylation was found in TEs on chrX of TS, which was essential for inhibiting their activities and maintaining the stability of the chromosome. Previous studies indicated that the insertion of mobile elements into the early embryo can disrupt genes, leading to diseases [23,24]. Therefore, it would be interesting to consider the methylation features of TEs in the villi during early spontaneous abortions [25,26]. We speculate that the regulation of methylation may be a self-rescue process for live TS fetuses in order to maintain the stability of the genome and their survival. There may, therefore, be significant differences in the genome methylation levels between the survived infants and spontaneous abortions in patients with TS. This is an area that still needs to be studied further by RNA expression profiling and differentially methylated regions analysis.

In summary, based on the characteristics of the methylation spectra of different regions of chrX, it is speculated that TS individuals can only regulate the gene expression on the X chromosome by DNA methylation in order to survive.

### 4.3. Differentially Methylated Genes (DMGs) and Functional Enrichment Analysis

We identified 157 genes which were differentially methylated among TS 46XY and 46XX individuals. The pathway and process enrichment analyses showed that the enriched terms included mitophagy, tubulin deacetylation, regulation of DNA-binding transcription factor activity and protein autophosphorylation. This indicated that in order to compensate for the defects caused by the loss of sex chromosomes, TS individuals could introduce epigenetic regulation by differential methylation of certain related genes, thereby overcoming the deleterious effects and allow their survival. Therefore, these DMGs and enriched pathways may have important implications for the survival and development of patients with TS. As an example, one of the most significant pathways identified from TS DMGs (hypomethylated) included the basal transcription factors, ko03022, TAF1, TAF9B, and TAF7L, which are responsible for the regulation of cellular transcription. Additionally, one of the enriched terms refers to regulation of DNA-binding transcription factor activity (GO:0051090) (AR, FLNA, IRAK1, TAF1, FZD1, IKBKG, CCDC22, BEX1, and EIF2AK4), which also suggest these factors may play important roles in several regulatory events. The ARHGAP6 gene was identified as a hypomethylated gene in TS 46X0 when compared to 46XX, and this was also detected by previous studies [3,10]. It was reported to undergo X inactivation and its expression was up regulated in TS.

We also identified six potential candidate genes which were hypermethylated when comparing TS with 46XX (BCOR, AK2, and RP11-453F18_B.1) and TS with 46XY (HOXB3, HOXB-AS3, and DTX2). Further investigation of these genes in previous studies showed that RP11-453F18_B.1 (Firre-hnRNPU) is involved in a significant biologically relevant interaction, and hnRNPU has previously been shown to have a role in XIST localization [27].

With respect to AK2 (adenylate kinase 2, which is localized in the mitochondria), this enzyme catalyzes the reversible transfer of the terminal phosphate group between ATP and AMP and plays an important role in cellular energy homeostasis and in adenine nucleotide metabolism. It is expressed at high levels in the heart, liver, and kidneys, and is associated with a fatal form of severe combined immunodeficiency. This condition is characterized by absence of granulocytes and an almost complete deficiency of lymphocytes in peripheral blood. There is also hypoplasia of the thymus and secondary lymphoid organs, and a lack of innate and adaptive humoral and cellular immunity, leading to fatal septicemia within days after birth.

BCOR was also found to be hypermethylated. BCOR is a transcriptional corepressor, and it may specifically inhibit gene expression when it is recruited to the promoter regions by sequence-specific DNA-binding proteins such as BCL6 and MLLT3 [28]. It regulates the functions of mesenchymal stem cells by epigenetic mechanisms [29]. Altogether, the hypermethylation of these three genes may play important roles, and we speculate that TS individuals may have a tightly regulated gene expression model with highly efficient methylation of certain related genes.

## 5. Conclusions

In conclusion, the methylation profile in patients with TS may be determined by the number of X chromosomes, and the patterns of methylation were found to be associated with the maintenance of genomic stability and an improvement of gene expression. Differential methylation regions and candidate genes may also be associated with the regulation of methylation within the whole genome. The discovery of DMGs and their accompanying pathways might reveal further potential epigenetic modulations of DNA methylation in TS, and this will lead to a better understanding of this complicated condition both scientifically and clinically.

## Figures and Tables

**Figure 1 jpm-12-00872-f001:**
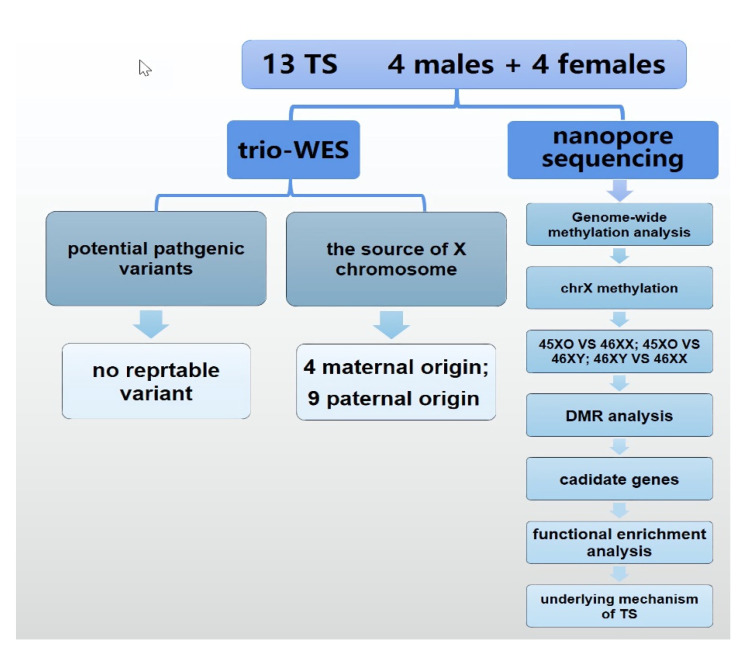
The objective and a flowchart of this study.

**Figure 2 jpm-12-00872-f002:**
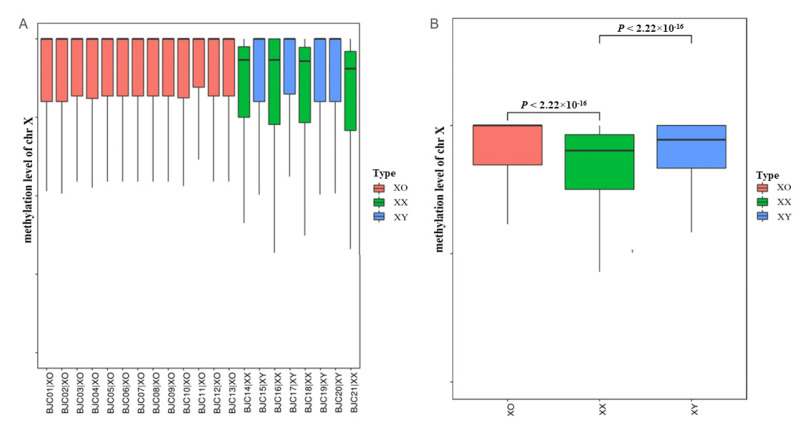
The methylation levels of 13 TS individuals and 8 healthy controls. Red boxes: TS, green boxes: female controls, and blue boxes: male controls. (**A**) The boxplots represent the medians and the 25th and 75th percentile methylation levels of each sample. (**B**) The statistical significance was tested using a two-sided Wilcoxon signed rank sum test. *p* values are indicated on the top of the boxes.

**Figure 3 jpm-12-00872-f003:**
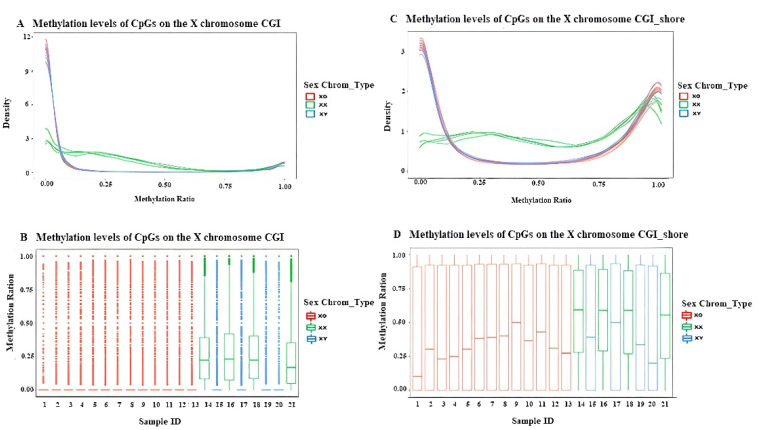
The methylation levels of 13 TS individuals and 8 healthy controls in the CGI and CGI_shore regions on chrX. The methylation density (**A**) and methylation ratio (**B**) of each sample in the CGI regions. The methylation density (**C**) and methylation ratio (**D**) of each sample in the CGI_shore regions. (Red: TS, green: female controls, and blue: male controls).

**Figure 4 jpm-12-00872-f004:**
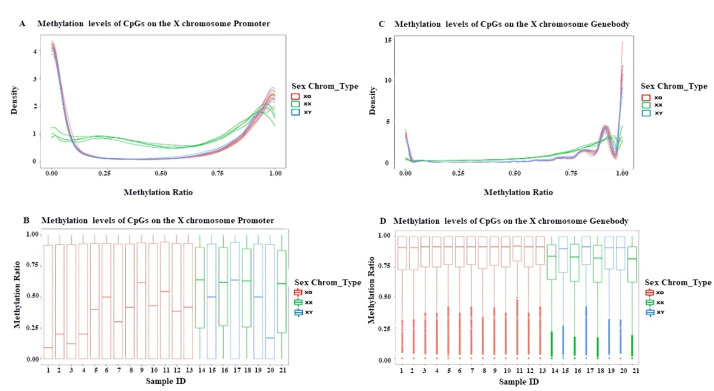
The methylation levels of 13 TS individuals and 8 healthy controls in the promoter and genebody regions on chrX. The methylation density (**A**) and methylation ratio (**B**) of each sample in the promoter regions. The methylation density (**C**) and methylation ratio (**D**) of each sample in the genebody regions. (Red: TS, green: female controls, and blue: male controls).

**Figure 5 jpm-12-00872-f005:**
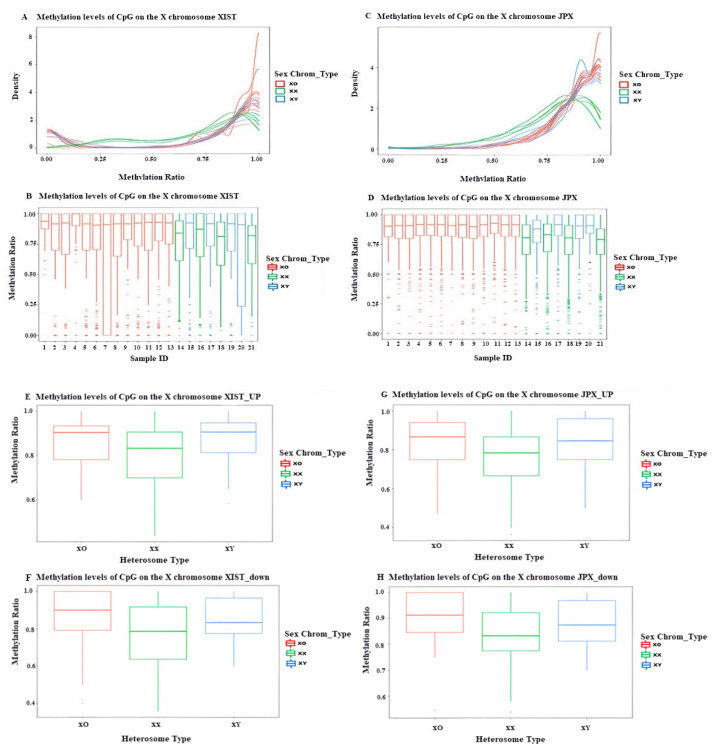
The methylation levels of 13 TS individuals and 8 healthy controls in XIST and JPX regions on chrX. The methylation density (**A**) and methylation ratio (**B**) of each sample in the XIST regions. The methylation density (**C**) and methylation ratio (**D**) of each sample in the JPX regions (**E**–**H**). The methylation levels of 13 TS individuals and 8 healthy controls in the 2K upstream and downstream of the transcription start sites of the XIST and JPX regions. (Red: TS, green: female controls, and blue: male controls).

**Figure 6 jpm-12-00872-f006:**
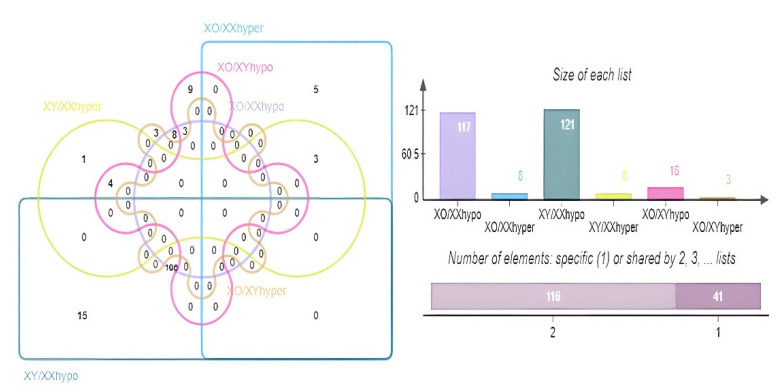
A Venn diagram of the differentially methylated (hypermethylated and hypomethylated) genes among the three groups of Turner syndrome 45X0 and controls 46XY and 46XX.

**Figure 7 jpm-12-00872-f007:**
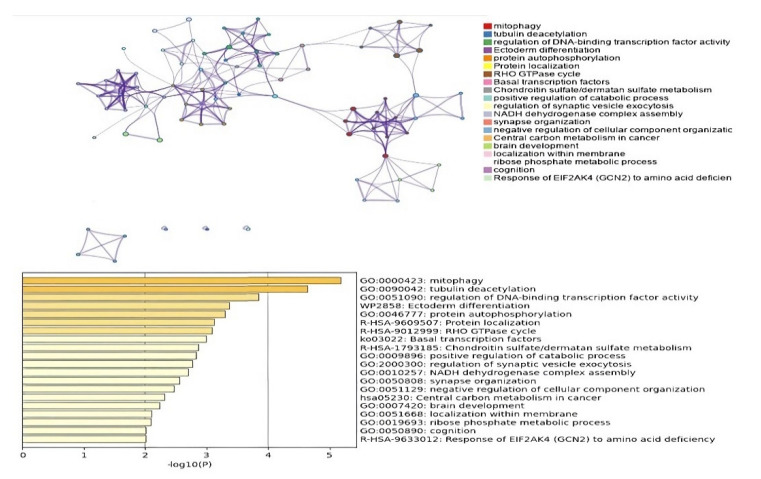
Functional enrichment analyses of the differentially methylated genes in 45X0 and 46XY compared to 46XX. An interactive network of the top 20 enriched terms colored by cluster ID, where nodes that share the same cluster ID are typically close to each other. The top 20 clusters with their representative enriched terms (one is given per cluster).

**Figure 8 jpm-12-00872-f008:**
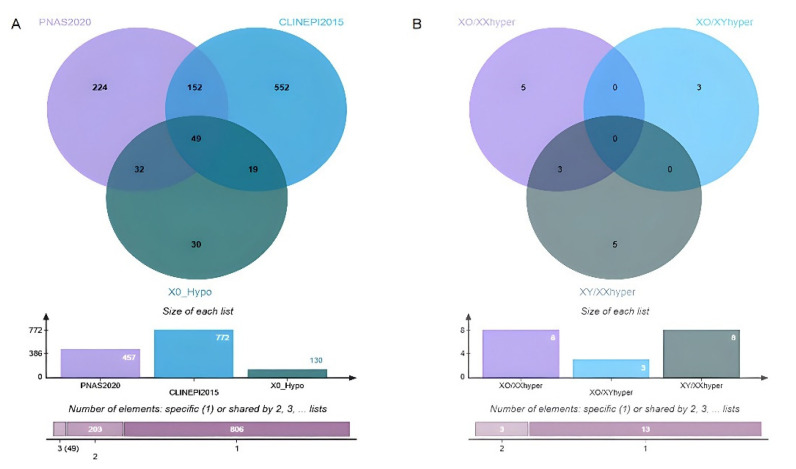
(**A**) A Venn diagram of the candidate genes of X0_Hypomethylated which were also detected as hypomethylated in TSvs46XX in a previous study. (**B**) A Venn diagram of the differentially hypermethylated genes among Turner syndrome 45X0 and controls 46XY and 46XX subjects.

## Data Availability

Individual level data cannot be made publicly available to protect the privacy of the participants. The datasets used and analyzed during the current study are available from the corresponding author on reasonable request.

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
