# Peer review of "DNA Hypermethylation and a Specific Methylation Spectrum on the X Chromosome in Turner Syndrome as Determined by Nanopore Sequencing"

_jpm, 2022, doi:10.3390/jpm12060872_

Round 1

Reviewer 1 Report

Please see the attached review.

Author Response

Response to Reviewer 1 Comments

Dear reviewer:

Thank you for your comments and providing good suggestions for my manuscript. We have revised my manuscript according to your suggestion. My answers to the comments point- by- point is as follow:

Point 1:I would recommend using a phrase “patients with TS” not “TS patients”.

Response 1: We have replaced all the “TS patients” with “patients with TS” in the manuscript, for example: L38,76,84,90,95,100,137,169,177.

Point 2: Line 16: “Trio-WES analysis failed to detect significant information.”- I would suggest adding more detailed information.

Response 2:We added more information as follows: Trio-WES and exome-based copy number variant (CNV) analyses were performed on WES data, we did not find any pathogenetic or likely pathogenic single-nucleotide variant (SNV), or small insertion/deletion, or chromone copy number variant related to typical clinical characteristics of TS, such as short stature, hypergonadotropic dysplasia, cardiac and renal malformation et al.

Point 3: Line 38: “variable phenotypes are related to the monosomy of chrX”- please explain an abbreviation. This part of the introduction should be widen, the mosaicism should be described. We are more aware now that probably a true monosomy X is very rare or does not exist postnatally, probably there is a form of tissue mosaicism that is not detected. Please add more references.

Response 3: We change “chrX” to “chromosome X(chrX)”, and added some contents about the mosaic in patients with TS. L41~44:

Highly variable phenotypes are related to the monosomy of chromosome X, the vari-ant absence of genetic material on the X chromosome, and some believed all 45X0 in-dividuals lived were cryptic mosaic, mosaic in all organs or tissues might be contrib-uted, but it is really difficult to conclude in a living patient [2].

Point 4: Lines 268-271: “The characteristic clinical features of TS include dysmorphic features, short stature, cardiovascular, renal, skeletal, endocrine abnormalities and hearing problems. The highly variable clinical features, and the relationship between genotypes and phenotypes is still not well understood.”- this is unnecessary repeated from the Introduction.

Response 4: We deleted the sentences. L326~328.

Point 5: Line 344: “3. DMGs and functional enrichment analysis”- I would suggest using the whole phrase, not the abbreviation.

Response 5: We change DMGS to differentially methylated genes (DMGs).L423

Point 6: Line 385: “In conclusion, we conducted both second and third-generation sequencing on TS individuals.”- either widen or do not repeat this information.

Response 6: We deleted the sentences. L465

Point 7: line 392: “debilitating condition”- in my opinion it should be definitely changed! Women with TS can live a fulfilling life despite the genetic condition.

Response 7: we change the word “debilitating” to “complicated”.L472

Point 8:Unfortunately I cannot see the figures and tables in the supplementary materials

Response 8:We upload the supplementary materials again.

Reviewer 2 Report

This study describe a novel approach to investigate genetic and epigenetic abnormality in TS patients. Nanopore sequencing was combined with trio-WES and used for the first time, which revealed new information on top of previous studies using other methods. This study also performed in-depth investigation of epigenetic changes potentially caused by the loss of an X chromosome. The results suggested that to compensate the loss of an X chromosome, there may be hypomethylation and active transcription from the remaining X chromosome in TS patients, which seems to be reasonable. The authors also explained the discrepancy of findings between previous studies and theirs based on the difference in sequencing technology, which provides helpful information for the audience.

There are several minor comments.

    1. In normal female, there is X chromosome inactivation. Can the authors comment on the existence and extent of X chromosome inactivation in TS patients?
  1. The resolution of figures seems to be low, making it hard for me to take a closer look at the data.
  2. There are some typos and grammar mistakes that need to be corrected.
  3. The sentence 'Trio-WES analysis failed to detect significant information' in the abstract seems to be confusing. I would suggest the authors clarify what you mean by 'significant information'. Otherwise, readers may think that there are no interesting findings in this study at all.
  4. Can the authors comment on how the information collected in this study contribute to the treatment and management of TS patients, if any? Why is this study important to the medical professional in this field?

Author Response

Response to Reviewer 2 Comments

Dear reviewer:

Thank you for your comments and providing good suggestions for my manuscript. We have revised my manuscript according to your suggestion. My answers to the comments point- by- point is as follow:

Point 1:In normal female, there is X chromosome inactivation. Can the authors comment on the existence and extent of X chromosome inactivation in TS patients?

Response 1: Thank you for your question. Maybe I did not express clearly, what I want to expressed is that inactivated X chromosome in normal female patients was different from X chromone in TS, in fact, X chromosome in TS was maintain as much activation as possible, in order to keep only X chromosome, work as expected. 

According to the previous studies, X inactive specific transcript (XIST) region of X chromone determined the activity of chrX. in normal female, one of the X chromosomes was inactive with hypermethylation of XIST region. In patients with TS, who has only one X chromosome, chrX was always activated in order to keep the genome balance and the individual alive. In 45,X0 patients, hypermethylation existed in XIST, average methylation levels and patterns were similar to 46,XY. But in some TS patients who have two X chromosome, and one of X chromosome was structurally rearranged, their abnormal X will be preferential inactivated, which can restore a balance in genome. In mosaic TS, for example 45X0/46XX,it’s reported that the inactivation of X chromosome may be non-random in mosaic TS, but the mechanism is not clear, which may be related to the cryptic abnormal structure of X chromosome, a cryptic structural aberration not identified by conventional G-banding or even by high-resolution banding. Such cryptic aberrations induce chromosomal instability and give rise to 45,X cell lines, along with other X chromosome aberrations, such as microdeletions and fragmentary marker chromosomes(PMID: 7762970).

Point 2: The resolution of figures seems to be low, making it hard for me to take a closer look at the data.

Response 2:Thank you for your comment, we checked and increased the resolution of figures.

Point 3: There are some typos and grammar mistakes that need to be corrected.

Response 3: Thank you for your comment, we corrected the grammar mistakes, and ask native speaker to edited the manuscript.

Point 4: The sentence 'Trio-WES analysis failed to detect significant information' in the abstract seems to be confusing. I would suggest the authors clarify what you mean by 'significant information'. Otherwise, readers may think that there are no interesting findings in this study at all.

Response 4: Thank you for your comment, we added more information as follows: Trio-WES and exome-based copy number variant (CNV) analyses were performed on WES data, we did not find any pathogenetic or likely pathogenic single-nucleotide variant (SNV), or small insertion/deletion, or chromone copy number variant related to typical clinical characteristics of TS, such as short stature, hypergonadotropic dysplasia, cardiac and renal malformation et al. abstract L16~17, discussion: L341~346

Reviewer 3 Report

The manuscript by Fan and co-authors “DNA hypermethylation and a specific methylation spectrum 2 on the X chromosome in Turner syndrome as determined by 3 nanopore sequencing» addresses the issue of genetic and epigenetic causes of Turner syndrome. This is of considerable interest for both fundamental biology and practical medicine. The authors combine trio-WES and Nanopore sequencing to separate the effects of rare genetic variants from the DNA methylation-based gene regulation. They managed to identify a set of differentially methylated genes, that distinguish 45X0 TS individuals from healthy donors and can be potentially involved in the development of associated pathogenic conditions. However, before being accepted for publication, some points should be addressed.

Major:

1) when comparing methylation profiles of 46XX to 46XY and 45XY samples by Nanopore sequencing, is in possible to distinguish the impact of maternal and paternal chromosomes on the overall methylation patterns? I’d rather include and discuss these data.

2) DNA from blood samples is easy to obtain, but the majority of abnormalities associated with TS involve other organs and tissues, which methylation profiles are obviously different from that of blood cells. To what extent the conclusions concerning the absence of differences in methylation levels and patterns in autosomes can be generalized to the whole organisms and, in particular, organs affected?

3) Apart form PAR region, do you see a particular methylation pattern for X-linked genes that escape silencing?

4) The authors try to explain why their data are so different for those obtained by Zhang, X et al [2] and Trolle et al [16] through comparison of methylation detection approaches used. However, they merely describe the nanopore sequencing basics without real comparison of advantages and possible pitfalls of the method.

5) Sample preparation for trio-WES should be described in more detail.

6) Figure legends and axes labeling are barely legible! The font size should be increased.

Minor points:

The manuscript requires thorough proofreading! For example:

84: healthy female caryotype should be 46XX ?

87: whole exon should be whole exome?

105: «fragmented» instead of «interrupted»?

106: addition of a reaction of long DNA fragments ???

280 “Initial analysis based on trio-WES indicated that the original of chrX in four TS patients 280 was maternally-derived, ...» “origina” or “origin”? In both cases, it is not comatible with “was maternally-derived”. Please, rephrase.

288 “2.1 Hypermethylation of genome-wide DNA methylation in TS”. Are you studying hypermethylation of methylation? Please, rephrase.

326 “conducive” – “conductive”?

351 “Therefore, these DMGs and enriched pathways and may have important implications for survival and development of TS patients. “ - remove “and”?

365 “Further investigation of these genes in previous stud ies showed that, RP11-453F18_B.1, (Firre-hnRNPU is a biologically relevant interaction and hnRNPU has previously been shown to have a role in XIST localization)[24] – check punctuation.

Author Response

Response to Reviewer 3 Comments

Dear Editors and reviewers:

Thank you for your comments and providing good suggestions for my manuscript. We have revised my manuscript according to your and reviewer's suggestion. My answers to the comments point- by- point is as follow:

Major:

Point 1:when comparing methylation profiles of 46XX to 46XY and 45XY samples by Nanopore sequencing, is in possible to distinguish the impact of maternal and paternal chromosomes on the overall methylation patterns? I’d rather include and discuss these data.

Response 1:In 13 patients, four patients (P2,P5,P7,P13) were maternal-derive, and the rest of 9 patients were paternal-derive, but there was no significant difference in the methylation profiles between the two groups. As we can see in Figure 2-A(in the following figure), red boxes represented all the TS patients. In the future, we will expand the sample size for observation and comparison in maternal and paternal X chromone in TS. See L198~202.

figure 2 in the PDF file

Point 2:DNA from blood samples is easy to obtain, but the majority of abnormalities associated with TS involve other organs and tissues, which methylation profiles are obviously different from that of blood cells. To what extent the conclusions concerning the absence of differences in methylation levels and patterns in autosomes can be generalized to the whole organisms and, in particular, organs affected?

Response 2: Thank you for your comments. In this study, we selected the patients whose karyotype of white cells in peripheral blood was 45,X, and try the best to exclude mosaicism from the general karyotype by counting metaphase chromosomes (≥50), which was the major way. And deep trio-whole exome sequencing(average depth of sequencing is 240x, the minimum depth is more than 160x)in TS individuals, also contributed to recognize mosaic to some extent. But the detection of mosaicism in human disease has been challenging. There is a hypothesis all the TS individuals alive were cryptic mosaic, the ratios of abnormal cells in different tissues of variant karyotype were not the same. And it is reported that 50% TS had ≥ 5% difference in the degree of mosaicism between the left- and right-hand sides of the buccal mucosa (PMID:31466136), on the other hand, the proportion of 46,XX cells increases with time in women with Turner syndrome from a 10-year follow-up study (PMID: 25587646). So, It is really difficult to identify how whole organisms’ methylation change in TS by peripheral blood DNA. Now we still cannot exactly come to conclude,that what extent the methylation levels and patterns in autosomes can be generalized to the whole organisms. In future study, analysis of tissues specific methylation profile combine with peripheral blood methylation profile at the same time might be helpful to better understand.

Point 3:Apart form PAR region, do you see a particular methylation pattern for X-linked genes that escape silencing?

Response 3: Apart from PAR region, there were total of 125 differentially methylated genes on X chromosome were identified in 45,X0 compared to 46XX,117 of which were significantly differential hypomethylated, and the methylation pattern increased the expression of the genes. Besides there were 106 common genes in X0/XX and XY/XX hypo (Table S2). For example, hypomethylated genes about basal transcription factors, ko03022, TAF1, TAF9B and TAF7L, which are responsible for the regulation of cellular transcription. Additionally, regulation of DNA-binding transcription factor activity (GO:0051090) (FLNA, IRAK1, TAF1, FZD1, IKBKG, CCDC22, BEX1 and EIF2AK4), which also suggest these factors may play important roles in the regulatory events. (See L424~441)

Point 4: The authors try to explain why their data are so different for those obtained by Zhang, X et al [2] and Trolle et al [16] through comparison of methylation detection approaches used. However, they merely describe the nanopore sequencing basics without real comparison of advantages and possible pitfalls of the method.

Response 4:We added information about the advantages and advantages of the method in the discussion. : The significant advantages of nanopores sequencing include no labeling, long-reading, and will not be affected by the amplification bias introduced by PCR, and the number of CpG sites obtained is more than that of the traditional methylation detection method, so nanopore sequencing can faithfully estimate DNA methylation land-scopes. But nanopore sequencing does have its disadvantages. It tends to be error prone, error rates in nanopore sequencing can be about 15%. This error rate can be tolerable by sequencing large amounts of the same sequence. Line 373-380.

Point 5: Sample preparation for trio-WES should be described in more detail.

Response 5: We added more details about trio-WES in the manuscript. L100~130.

Point 6: Figure legends and axes labeling are barely legible! The font size should be increased.

Response 6: We will edit the figures again followed the instruction of the journal.

Minor points:

Point 7:  84: healthy female caryotype should be 46XX ?

Response 7: We corrected “4 males (46XY) and 4 females (46XY)” to “4 males (46XY) and 4 females (46XX)”. L92

Point 8:  87: whole exon should be whole exome?

Response 8:  We corrected exon to exome.L96

Point 9: 105: «fragmented» instead of «interrupted»?

Response 9: We corrected interrupted to fragmented.L138

Point 10: 106: addition of a reaction of long DNA fragments???

Response 10: We change the sentence:  For Nanopore sequencing, library preparations were done using the ligation sequencing kit (Cat# SQK-LSK109, Oxford Nanopore Technologies). All procedures were preformed according to the manufacturer’s protocols. L139~142.

Point 11: 280 “Initial analysis based on trio-WES indicated that the original of chrX in four TS patients 280 was maternally-derived, ...» “origina” or “origin”? In both cases, it is not comatible with “was maternally-derived”. Please, rephrase.

Response 11: “origina” should be “origin”, and we rephrase the sentence as follow: Initial analysis based on trio-WES indicated, that the X chromones in four patients with TS were maternally-derived, whereas the other nine were paternally-derived, which is consistent with previous studies, most TS lost paternal X chromone. L338

Point 12 :  288 “2.1 Hypermethylation of genome-wide DNA methylation in TS”. Are you studying hypermethylation of methylation? Please, rephrase.

Response 12: we change the subtitle to “Genome-wide DNA methylation in TS”.L352

Point 13:326 “conducive” – “conductive”?

Response 13: Thank you for your careful work regarding our manuscript. It should be “conductive”, we have corrected. L403

Point 14:351 “Therefore, these DMGs and enriched pathways and may have important implications for survival and development of TS patients. “ - remove “and”?

Response 14: The second “and” should be removed. L431

Point  15:365 “Further investigation of these genes in previous stud ies showed that, RP11-453F18_B.1, (Firre-hnRNPU is a biologically relevant interaction and hnRNPU has previously been shown to have a role in XIST localization)[24] – check punctuation.

Response 15 : we check the punctuation, and corrected as fowllow:

Further investigation of these genes in previous studies showed, that RP11-453F18_B.1 (Firre-hnRNPU) is a biologically relevant interaction, and hnRNPU has previously been shown to have a role in XIST localization.L444~447.

Reviewer 4 Report

I have reviewed the manuscript “DNA hypermethylation and a specific methylation spectrum on the X chromosome in Turner syndrome as determined by nanopore sequencing” by Fan et al submitted for publication in the Journal of Personalized medicine (MDPI).

My comments and questions to the authors are given below:

In the study, the authors aimed to investigate the DNA methylation spectrum in the Chr. X in the Turner syndrome (TS) patients to understand the epigenetic modulation and its impact on the clinical pathology of TS. In this study, a total of 13 TS patients and 8 age-matched control subjects were recruited and their blood DNA samples were analyzed by Trio-WES and nanopore sequencing methods to compare and analyze the DNA methylation pattern. The authors tried to find any novel epigenetic signature among TS patients that could identify novel differentially methylated genes (DMGs) and their role in the clinical pathology of TS.

The overall study design and methods are properly designed. But has some major concerns to address.

  1. All the figures, (Figures 1 to 8) in the manuscript are not clear, I tried to zoom in and see the scripts in the figures, and they are not legible! I recommend redrawing the figures or uploading a high-quality picture with better quality, so they don’t get smudged when we zoom them!
  2. I see they listed Supplementary Materials (Figures S1 to S5 and Tables S1 to S4) and the legend in the manuscript but I do not see them available for download. I want to see those supplemental figures and tables, plan to upload them to my reviewer portal.

Questions:

  1. About the study samples, the eight healthy controls (4 males 46XY and 4 females 46XX), are they random controls or family relatives to the patients like siblings? Also, do all the parents of the 13 TS patients included in the Trio-WES study?
  2. Are the TS patients and the control participants from the same ethnic group?
  3. Are the TS patients were screened for mosaicism? Since the authors mentioned the varied clinical features among the 13 study TS patients, I could not see their phenotype data since I do not have Table S1.
  4. The authors mentioned that among 13 TS patients in this study 4 patients were carrying maternal ChrX and 9 were carrying paternal ChrX. Do the authors compare the phenotype data of these two groups to find out do they share any similar TS phenotypes and does the methylation profile differs among these two groups of patients?

Comments:

The research methods of the study are well designed, but due to the poor quality of the figures and lacking supplemental figures and tables, I cannot evaluate the argument the authors put forth in this manuscript. I look forward to seeing the clear figures and supplemental materials to provide my final comments.

Author Response

Response to Reviewer 4 Comments

Dear reviewer:

Thank you for your comments and constructive suggestions regarding our manuscript. We have revised my manuscript according to your suggestion. My answers to the comments point- by- point is as follow:

Point 1: All the figures, (Figures 1 to 8) in the manuscript are not clear, I tried to zoom in and see the scripts in the figures, and they are not legible! I recommend redrawing the figures or uploading a high-quality picture with better quality, so they don’t get smudged when we zoom them!

Response1: We checked and increased the resolution of figures follow the instruction of the journal.

Point  2: I see they listed Supplementary Materials (Figures S1 to S5 and Tables S1 to S4) and the legend in the manuscript but I do not see them available for download. I want to see those supplemental figures and tables, plan to upload them to my reviewer portal.

Response 2: We will carefully upload the supplementary materials again.

Questions:

Point 3: About the study samples, the eight healthy controls (4 males 46XY and 4 females 46XX), are they random controls or family relatives to the patients like siblings? Also, do all the parents of the 13 TS patients included in the Trio-WES study?

Response 3: The healthy controls are random controls, and all the patients with TS and their parents underwent Trio-WES study.

Point 4: Are the TS patients and the control participants from the same ethnic group?

Response 4: All the controls and TS patients are Han ethnic, which is biggest nationality in China.

Point 5: Are the TS patients were screened for mosaicism? Since the authors mentioned the varied clinical features among the 13 study TS patients, I could not see their phenotype data since I do not have Table S1.

Response 5: I am sorry for the absence of table S1, we upload the supplements again, and thank you for your carefully review. In this study, we selected the patients whose karyotype of white cells in peripheral blood was 45,X, and try the best to exclude mosaicism from the general karyotype by counting metaphase chromosomes (≥50), which was the major method. And deep trio-whole exome sequencing(average depth of sequencing is 240x, the minimum depth is more than 160x)in TS individuals, also contributed to recognize mosaic. But still the detection of mosaicism in human disease has been challenging,especially in TS. There is a hypothesis that all the TS individuals alive were cryptic mosaic, the ratios of abnormal cells in different tissues of the same TS patients were not the same, even varied between age. 

Point 6:The authors mentioned that among 13 TS patients in this study 4 patients were carrying maternal ChrX and 9 were carrying paternal ChrX. Do the authors compare the phenotype data of these two groups to find out do they share any similar TS phenotypes and does the methylation profile differs among these two groups of patients? 

Response 6: In 13 patients with TS, four patients (P2,P5,P7,P13) were maternal-derive, and the rest of 9 patients were paternal-derive. Compare the phenotype of these two groups (Table S1), they shared the same phenotypes of short stature and ovarian hypoplasia, all the patients had short stature and ovarian hypoplasia. But there was not prominent difference in the incidence of other systematic complications, a minor discover that all of four maternal-derive patients had epicanthal folds, shield chest, cubitus valgus, but the other group only one or three patients (1/9,3/9) had these phenotypes.

We have compared the methylation profiles of maternal and paternal X in TS, but there was no significant difference in the methylation profiles between the two groups. As we can see in Figure 2-A(in the following figure), red boxs represent TS patients. See L376~379. We did not find any prominent pattern or difference in patients between paternal or maternal chrX. The other possible reasons maybe due to both of TS have only one chrX, the activation and gene expression of X chromosome will be modulated as closely as possible to diploid state, the regulation may reduce the gap.  See L198~202

Figure 2 in the PDF files.

Point 7: Comments: The research methods of the study are well designed, but due to the poor quality of the figures and lacking supplemental figures and tables, I cannot evaluate the argument the authors put forth in this manuscript. I look forward to seeing the clear figures and supplemental materials to provide my final comments.

Response 7:  Thank you very much for your comments, we will update the figure and upload the supplementary materials again.

Round 2

Reviewer 3 Report

In its current shape the manuscript by Fan and co-authors demonstartes some improvements, hovewer, not all my points were properly addressed. In the first place, I am still not satisfied with the figures, the readability of labels is still poor, at least in the PDF I have an access to. Next, I'd suggest to incorporate the reply to my point 2 into the discussion section.

Author Response

               Response to Reviewer 3 Comments

Dear Editors and reviewers:

Thank you for your comments and providing good suggestions for my manuscript. We have revised my manuscript according to your and reviewer's suggestion. My answers to the comments point- by- point is as follow:

Point 1:In its current shape the manuscript by Fan and co-authors demonstartes some improvements, hovewer, not all my points were properly addressed. In the first place, I am still not satisfied with the figures, the readability of labels is still poor, at least in the PDF I have an access to. Next, I'd suggest to incorporate the reply to my point 2 into the discussion section.

Response 1: Thank you for your comments. We have modified picture again, and make the label clearer. Also, we added the point 2 into the discussion.L305~310 and L326~338.
